# Analytical Performance of Different Laboratory Methods for Measuring Susoctocog-Alfa

**DOI:** 10.3390/diagnostics12081999

**Published:** 2022-08-18

**Authors:** Cristina Novembrino, Ilaria Quaglia, Angelo Claudio Molinari, Alessandra Borchiellini, Antonio Coppola, Rita Carlotta Santoro, Massimo Boscolo-Anzoletti, Eleonora Galbiati, Ezio Zanon, Alessandra Valpreda

**Affiliations:** 1Fondazione IRCCS Ca’ Granda Ospedale Maggiore Policlinico, Angelo Bianchi Bonomi Hemophilia and Thrombosis Center, 20122 Milan, Italy; 2Center for Thrombosis and Hemorrhagic Diseases, IRCCS Humanitas Research Hospital Rozzano, 20089 Milan, Italy; 3Regional Reference Center for Inherited Bleeding Disorders, Istituto Giannina Gaslini, 16147 Genova, Italy; 4Regional Reference Center of Thrombotic and Hemorrhagic Disorders of the Adults, Department of Oncology and Hematology “Città della Salute e della Scienza”, 10100 Turin, Italy; 5Regional Reference Center for Inherited Bleeding Disorders, University Hospital of Parma, 43125 Parma, Italy; 6Hemostasis and Thrombosis Unit, Regional Reference Centre for Hemophilia, and Inherited Bleeding Disorders, AOPC, 88100 Catanzaro, Italy; 7Haemophilia Centre-General Medicine, Padua University Hospital, 35128 Padua, Italy; 8Regional Reference Center of Thrombotic and Hemorrhagic Disorders of the Adults, Laboratory Department “Città della Salute e della Scienza”, 10100 Turin, Italy

**Keywords:** acquired haemophilia A, recombinant porcine FVIII, assay methods

## Abstract

Recombinant porcine factor VIII (rpFVIII) is indicated for treating bleeding episodes in acquired haemophilia A, but there are few data regarding laboratory methods to adequately monitor treatment. This study involving three Italian laboratories aimed to evaluate the analytical performance of different assays for measuring rpFVIII. Five spiked rpFVIII samples (0.5–1.5 IU/mL) were analysed on three days, in triplicate, with eleven combinations of reagents (Werfen, Boston, MA, USA: SynthasIL and SynthaFax for one-stage assay, Chromogenix Coamatic FVIII for chromogenic assay), FVIII depleted plasmas (with or without von Willebrand factor—VWF) and calibrators (HemosIL human calibrator plasma, porcine calibrator diluted in FVIII deficient plasma with or without VWF). The assays were performed on ACL TOP analysers (Werfen, Boston, MA, USA). Intra- and inter-assay and inter-laboratory Coefficient of Variation (CV%) were calculated together with percentage of recovery (% recovery) on the expected value. The results showed that the reagent combinations reaching satisfactory analytical performance are: SynthasIL/human calibrator/deficient plasma+VWF (total recovery 99.4%, inter-laboratory CV 4.04%), SynthasIL/porcine calibrator/deficient plasma+VWF (total recovery 111%, inter-laboratory CV 2.75%) and Chromogenic/ porcine calibrator/deficient plasma+VWF (total recovery 96.6%, inter-laboratory CV 8.32%). This study highlights that the use of porcine standard (when available) and FVIII deficient plasma with VWF should be recommended.

## 1. Introduction

Acquired haemophilia A (AHA) is a rare bleeding disorder caused by the development of autoantibodies which neutralize the activity of endogenous coagulation factor VIII (FVIII) and/or accelerate its clearance. AHA occurs both in males and females with a previously normal haemostasis [1,2]. Autoantibody development is idiopathic in about 50% of cases, while in the remaining ones it is associated with autoimmune disorders, cancer, infections, drugs, or other triggering conditions [3,4]. Morbidity and mortality associated with AHA are high, especially in elderly patients with severe co-morbidities. International guidelines recommend treating bleeding caused by AHA as soon as possible in first-line therapy with bypassing agents, i.e., activated prothrombin complex concentrate (aPCC) or activated recombinant FVII (rFVIIa), or with susoctocog-alfa, a recombinant porcine FVIII (rpFVIII) [5,6]. Plasma-derived FVIII (pdFVIII) concentrates can also be an option for patients at high thromboembolic risk, especially in the presence of low-titre inhibitors (≤5.0 BU/mL), even if this is not essential, as reported in a previous Italian study [7]. rpFVIII has been suggested [8] even in patients with severe comorbidities as no cases of drug-related thrombosis have been reported in the literature.

Porcine FVIII (pFVIII) was introduced as a first-line therapy to control bleeding in AHA-related illnesses in the 1980s, especially in patients with high-titre inhibitors [9], thanks to the reduced cross-reactivity with human autoantibodies, which was significantly lower than with alloantibodies [10]. Anamnestic responses were infrequent, and only 20% of patients developed specific anti-porcine antibodies [11]. However, considerable side effects, such as chills, fever, headache, thrombocytopenia and uncommon anaphylactic responses were frequently reported until a new highly purified porcine FVIII (Hyate:C^®^, Porton Speywood Ltd., Wrexham, UK) was obtained [12].

Hyate:C^®^ was available on the market until 2004 when it was removed due to viral safety concerns and platelet hyper-aggregation [12,13,14,15]

In 2016, susoctocog-alfa (Obizur^®^, Baxalta US Inc.-Takeda company-, Bannockburn, IL, USA), a new recombinant B-domain-deleted FVIII, presenting porcine-sequence (rpFVIII), produced in a well-characterized baby hamster kidney (BHK) cell-line, and manufactured using two viral clearance steps to reduce the risk of potential pathogen transmission, was licensed in Italy. Phase 1 and Phase 2 studies with rpFVIII demonstrated comparable pharmacokinetic parameters to the pdFVIII concentrates [16] and a positive safety profile with efficacy in treating bleeds in patients with congenital haemophilia A [17,18] with inhibitors, as well as in patients with AHA [19,20,21].

It is recommended that in the case of congenital haemophilia, in which an accurate measurement of the FVIII level during replacement therapy is needed to evaluate the efficacy of treatment [22], and for AHA, a strict monitoring of rpFVIII should be performed [6].

The correct measurement of FVIII activity in AHA patients treated with Obizur^®^ is very important for patient safety and even for the correct use of resources. Recently, the “United Kingdom Haemophilia Centre Doctors’ Organisation” (UKHCDO) published recommendations for measuring FVIII in AHA patients in treatment with rpFVIII, excluding the use of the chromogenic assay (CSA) due to its high underestimation of FVIII levels except in the case of patients with AHA who receive emicizumab; however, the latter eventuality is still an off-label treatment [23].

An interesting study performed in an Italian laboratory investigated the differences in rpFVIII concentration, in vivo activity levels and in anti rpFVIII inhibitor titration, using aPTT reagents with different activators (ellagic acid and silica) in the one-stage assay (OSA). The rpFVIII level was dosed both on a human FVIII reference curve and on a porcine FVIII reference curve, resulting in it being strongly overestimated compared to what was expected by silica reagent and human calibrator [24].

On this basis, it is therefore still debated which methods and reagents may provide the best analytical performance to measure the level and plasma activity of Obizur^®^, in order to guarantee the correct management of all AHA patients treated with this drug.

The aim of this presented study was to evaluate the analytical performance of different combinations of reagents and calibrators for measuring rpFVIII activity in plasma.

## 2. Materials and Methods

This study involves three Italian laboratories (two in Milan, one in Turin) in which recombinant porcine FVIII (Obizur^®^) was measured using different assays (OSA and CSA).

### 2.1. Sample Preparation

All the study samples were prepared in one of the three laboratories by adding rpFVIII standard at a known concentration in commercial FVIII deficient plasma (HemosIL FVIII deficient plasma with von Willebrand factor-Werfen, Boston, MA, USA) to obtain five different rpFVIII concentrations: 0.05, 0.25, 0.50, 1.0 and 1.5 IU/mL, respectively. All the aliquots obtained were immediately frozen in liquid nitrogen and stored at −80 °C until the measurement.

### 2.2. Reagents

The samples were analysed with both the one-stage assay (OSA) and the chromogenic assay (CSA), which were performed with different reagents and calibrators. For OSA, the SynthasIL and the SynthaFax reagent (Werfen, Boston, MA, USA) with silica and ellagic acid, respectively, as activators were used, while the Chromogenix Coamatic FVIII kit (Werfen, Boston, MA, USA) was assayed for CSA. HemosIL human calibrator plasma (Werfen, Boston, MA, USA) and rpFVIII standard diluted in FVIII deficient plasma (HemosIL-Werfen, Boston, MA, USA) with and without von Willebrand Factor (VWF) were used to calibrate the OSA and CSA assays. All the assays were performed on an ACL TOP analyser (Werfen, Boston, MA, USA). The reagents were selected in order to include in the study the type of reagents representative of those mostly used in the routine laboratories for measuring FVIII:C (including chromogenic assay and one-stage assay based on aPTT reagents with both silica and ellagic acid as activators). All the tests were performed on the analyser according to the standard assay set up provided by the manufacturer.

### 2.3. Study Design

The study samples were distributed to the three different laboratories and measured using the same lots of reagents and calibrators. Each sample was tested with all possible combinations of reagents, deficient plasmas and calibrators. All the analyses were performed on three distinct days, in triplicate every day.

### 2.4. Statistical Analysis

Mean, Standard Deviation (SD) and Coefficient of Variation (CV%) were calculated in each of the three laboratories for the five samples analysed with all eleven methods to evaluate the intra- and inter-assay imprecision; inter-laboratory variability was also calculated and expressed as CV% for each of the assays evaluated. Analytical accuracy was determined for each method by calculating the percentage of recovery on the expected value. The difference between analytical performance has been statistically evaluated and the results inserted in the tables. Statistical analyses were performed by MedCalc(R) Statistical Software (MedCalc Software Ltd., Ostend, Belgium).

## 3. Results

Overall, a total of eleven combinations of reagents, deficient plasmas and calibrators were tested with OSA or CSA: (1) OSA: four combinations with SynthasIL (of which only three were tested by each of the laboratories); (2) OSA: four combinations with SynthaFax (of which only three were tested by each of the laboratories); (3) CSA: three combinations with Coamatic FVIII. In Table 1, inter- and intra-assay imprecision obtained for each of the three laboratories is shown. The reported CV% values are the mean of CV% obtained for the measures of the five samples at different rpFVIII concentrations. Almost all the methods exhibit repeatability and reproducibility of <10%. Some significant differences have been found between the imprecision of the three laboratories.

The results of analytical accuracy for each of the three laboratories are shown in Table 2. The reported values are the mean of % recovery obtained for the measures of the five samples at increasing concentrations of rpFVIII and assuming as acceptable a recovery for spiked samples ranging 70–130%. Despite some significant differences between laboratories, the SyntahsIL reagent gives accurate results when calibrated with both human and porcine standard provided that FVIII deficient plasma with VWF is used. An underestimation of rpFVIII activity was observed for all the combinations with the SynthaFax reagent, except one giving values higher than expected. The chromogenic assay instead showed adequate results only when calibrated with the porcine standard diluted in FVIII deficient plasma with VWF.

The overall results of the three laboratories are reported in Table 3 as total inter-laboratory CV% (mean of CV% obtained for all the measures of the five different samples in the three laboratories) and % recovery (mean of % recovery obtained for all the measures of the five samples in the three laboratories); for OSA, only the results obtained in at least two laboratories were reported. Considering both total imprecision (inter-laboratory variability) and accuracy, three of the eleven combinations of evaluated reagents showed satisfactory analytical performance.

The accuracy of these three combinations of reagents (SynthasIL with FVIII deficient plasma+VWF calibrated with human standard; SynthasIL calibrated with porcine standard in FVIII deficient plasma+VWF and Coamatic kit calibrated with porcine standard in FVIII deficient plasma+VWF) by comparing the values obtained for each of the five samples at different concentrations (0.05, 0.25, 0.50, 1.0, 1.5 IU/mL) with the expected values is graphically represented in Figure 1.

## 4. Discussion

The treatment of AHA is based on the eradication of the inhibitor and the management of bleeding diathesis [1,2]. Different studies showed that both bypassing agents, rFVIIa and aPCC, are effective [4,25] and safe [26]; nevertheless, particular attention in the use of these drugs in patients with evident thrombotic risk must always be maintained. Laboratory monitoring of treatment with bypassing agents is unfeasible using standard coagulation assays [27], representing a challenge in the management of AHA patients. On the other hand, rpFVIII is also effective in achieving the rapid control of bleeding in most AHA patients but can also be easily monitored using common standard FVIII assays [28]. The Obizur^®^ manufacturer recommends the use of the one-stage clotting assay to measure FVIII activity levels after Obizur^®^ dosing; however, the optimal methods and reagents to determine FVIII activity levels are not fully standardized. As a result, inter-laboratory variability in measurements can occur due to different instruments, methods of detection, assay set-up, reference standard calibration, reagent source and reagent composition. Generally, clinical laboratories use OSA to analyse post-infusion plasma samples collected from patients treated with rpFVIII, but CSAs are also in use and have variability resulting from different methods, instruments and assay kits [23,24]. The source and composition of aPTT reagents used for OSA was identified as a source of variability in results of post-infusion FVIII activity tests as well as potency assignment of rpFVIII [28]. The variability in FVIII activity measurements has clinical (and pharmacoeconomic) consequences, as FVIII levels are used to determine subsequent treatment protocols and dosing regimens. It is therefore important to identify which factors contribute towards discrepancies in results across a variety of assay systems with different reagents.

Most published data on patients treated with Obizur^®^ analysed the clinical outcomes and no information was reported regarding the method used to measure the recovery of the drug [29,30]. The discrepancies between OSA and CSA were instead highlighted in terms of reported methods [28,31,32]. Generally, no information regarding the calibrator was mentioned, except by Kruse-Jarres et al. [19] in their report, in which the World Health Organisation (WHO) human FVIII plasma standard as calibrator was described. Winikoff et al. [33] analysed the storage condition of plasma-derived porcine FVIII standard (Hyate:C) and the performance of the Bethesda assay for the determination of anti-porcine inhibitor titre; they reported similar results when the residual porcine FVIII activity was measured with OSA calibrated against human or porcine standard. This is partially in contrast with our data that show different results when the same reagent is calibrated with human or porcine standard, especially in the case of CSA.

The presented study tried to better clarify the analytical conditions underlying the discrepancies in the post-infusion monitoring of rpFVIII observed in different measurements with the OSA, with reagents containing silica and ellagic acid as activators, using deficient plasma with and without VWF and with the CSA, also in this case, on deficient plasma with and without VWF.

To our knowledge, this is the first study that highlights the importance of the presence of VWF in the FVIII deficient plasma, both for diluting standard rpFVIII and for performing OSA. In fact, our data show a high analytical accuracy with SyntahsiIL reagent when FVIII deficient plasma with VWF is used instead of FVIII deficient plasma without VWF. Conversely, a definitive conclusion is more difficult to draw in case of the reagent with ellagic acid as an activator (SynthaFAx) due to the high inter-laboratory variability found.

Our data furthermore demonstrate the possibility to reliably use the CSA when appropriately calibrated with porcine standard (preferably diluted in FVIII deficient plasma containing VWF). This finding could be useful in current practice because CSA is becoming the method of choice for monitoring FVIII replacement therapy in many laboratories [22] and particularly in specific clinical conditions due to the presence of interferences in patient plasma (such as anti-phospholipids antibodies, low dose heparin or a concomitant therapy with emicizumab). Recently, Hayden et al. [34] described the case of a patient with AHA treated first with Obizur^®^, and subsequently switched to emicizumab due to an unexpected diminishing half-life of rpFVIII. In this case, only the use of an appropriated CSA was useful to the management of the patient.

The role of emicizumab in AHA is in continuous evolution; in fact, the phase III AGEHA study was recently presented at the International Society on Thrombosis and Haemostasis (ISTH) Congress [35] whose encouraging results obtained on 12 patients made it possible to introduce emicizumab first in the treatment of Japanese patients with AHA. These data confirmed what was previously reported by Knoebl et al. [36], while Tiede et al. [37] had instead underlined the need to carry out further studies given the different characteristics of patients with AHA compared to those with congenital haemophilia, for whom the drug was initially designed. The monitoring of endogenous FVIII in patients with acquired haemophilia treated with this monoclonal antibody therefore becomes of primary importance for the laboratories in charge of measuring it. The AGEHA study involved monitoring endogenous factor VIII activity using a one-stage clotting assay with emicizumab neutralization by adding anti-emicizumab antibodies, but the researchers [38] showed that the chromogenic substrate assays using bovine or human coagulation factors also present a good correlation with the one-stage method and can be an alternative to measure the FVIII activity, although some attention must still be paid when choosing to use the human substrate.

## 5. Limitations

First, the reagents and coagulometers used in this study, although widespread, do not constitute the totality of the instrumentation and reagents available; consequently, some laboratory workers will only be able to obtain some indications from this work on how to organize themselves to deepen the topic. Secondly, the combinations of reagents with missing values in the analytical protocol were not analysed and not reported in the results section. Thirdly, these results, obtained on spiked plasma samples, should be confirmed in plasma from patients treated with rpFVIII, given that the rarity of AHA and the frequent emergency situations of diagnosis and treatment make it difficult to conduct formal studies and/or collect adequate amounts of plasma samples.

## 6. Conclusions

Our study shows that both OSA and CSA can be considered acceptable, repeatable and reproducible for measuring rpFVIII, but only if used under suitable analytical conditions. A correct calibration, and an adequate use of FVIII deficient plasma enriched or not with VWF, depending on the reagent and the method used, are of fundamental importance to be able to provide the clinician with the correct laboratory data, which is necessary to appropriately manage the patient with AHA being treated with Obizur^®^.

The study design, with the combinations of assays evaluated, addressed a complete investigation of the analytical conditions to measure rpFVIII activity in plasma, thus providing solid bases for further studies to confirm our results in real-world practice, collecting rpFVIII-treated patients’ plasma and using a wider set of reagents and analysers. All the reagents evaluated in the present study are commercially available, are used according to standard analytical conditions and, except for the rpFVIII calibrator, are not exclusively dedicated to rpFVIII measurements. All these features make our results reproducible in routine laboratories managing acquired haemophilia.

## Figures and Tables

**Figure 1 diagnostics-12-01999-f001:**
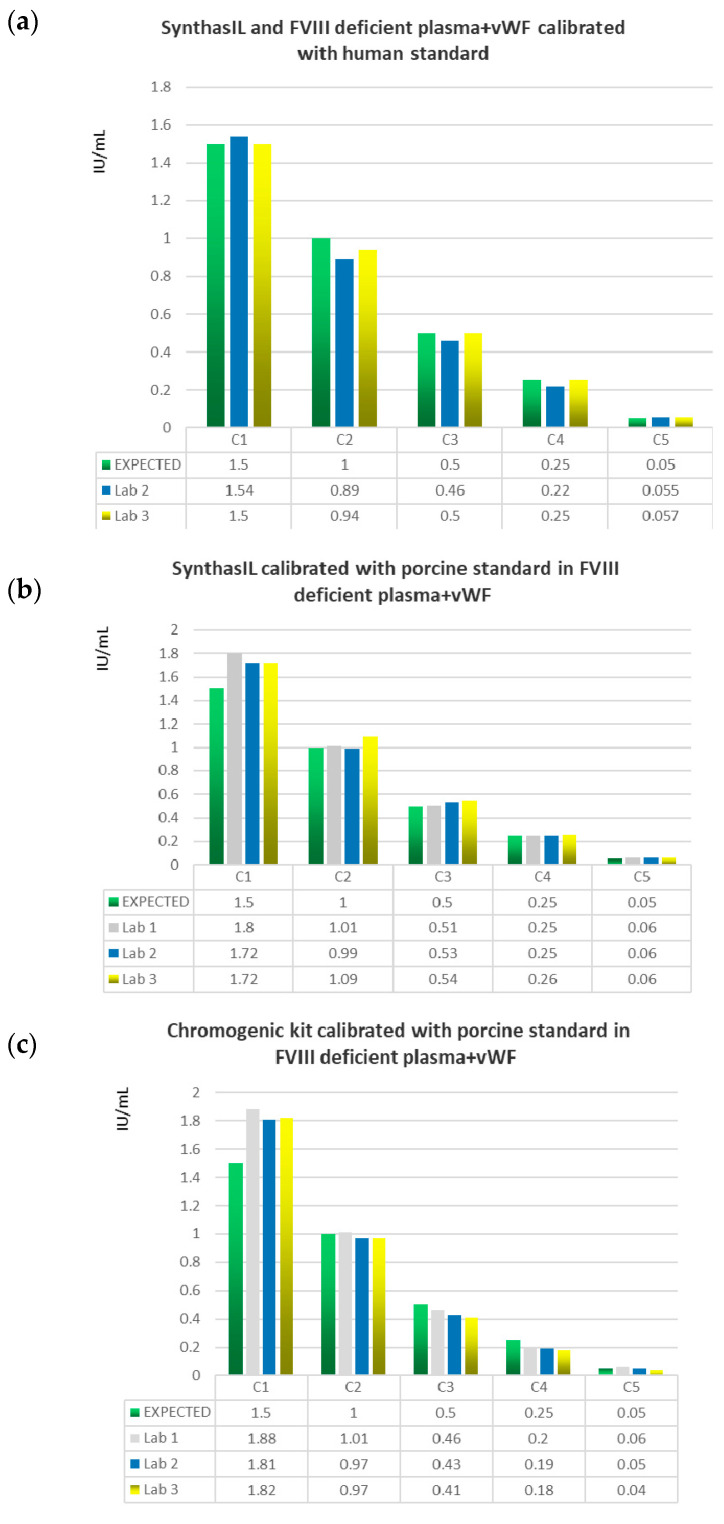
Histograms comparing the expected values of the five samples C1 = 1.5 IU/Ml, C2 = 1.0 IU/mL, C3 = 0.5 IU/mL, C4 = 0.25 IU/mL, C5 = 0.05 IU/mL) with the mean values obtained in each of the three laboratories with three different combinations of reagents: (**a**) SynthasIL with FVIII deficient plasma+VWF calibrated with human standard; (**b**) SynthasIL calibrated with porcine standard in FVIII deficient plasma+VWF; (**c**) Coamatic kit calibrated with porcine standard in FVIII deficient plasma+VWF.

**Table 1 diagnostics-12-01999-t001:** Intra- and inter-assay imprecision of all the methods used for rpFVIII measurement.

Method/Reagent	Intra-Assay CV %			Inter-Assay CV %		
	Lab 1	Lab 2	Lab 3	Lab 1	Lab 2	Lab 3
**OSA/SynthasIL** hs/FVIIIdef	3.2	-	-	5.6	-	-
**OSA/SynthasIL** hs/FVIIIdef+VWF	-	4.6	6.3	-	3.4	2.2
**OSA/SynthasIL** rpFVIIIs/FVIIIdef	2.8	1.8	2.3	3.4	1.7	2.3
**OSA/SynthasIL** rpFVIIIs/FVIIIdef+VWF	5.2	2.9	4.7	6.2	4.4	4.7
**OSA/SynthaFax** hs/FVIIIdef	-	2.7	-	-	2.2	-
**OSA/SynthaFax** hs/FVIIIdef+VWF	7.5	-	5.0	6.1	-	4.1
**OSA/SynthaFax** rpFVIIIs/FVIIIdef	3.4	1.9	2.2	3.4	2.2	2.5
**OSA/SynthaFax** rpFVIIIs/FVIIIdef+VWF	6.4	3.6	4.0	4.5	2.1	2.6
**CSA/Coamatic FVIII** hs	8.4 *	3.1 *	4.3	7.1	11.1	6.1
**CSA/Coamatic FVIII** rpFVIIIs in FVIIIdef	4.7	4.4	4.2	5.2	5.8	6.7
**CSA/Coamatic FVIII** rpFVIIIs in FVIIIdef+VWF	4.3	4.1	4.2	2.0 #§	7.6 #	6.8 §

* *p* = 0.01, # *p* = 0.001, § *p* = 0.002; Abbreviations: OSA = one-stage assay; CSA = chromogenic assay; hs = human standard; FVIIIdef = FVIII deficient plasma; FVIIIdef+VWF = FVIII deficient plasma with von Willebrand Factor; rpFVIIIs = recombinant FVIII porcine standard; CV = coefficient of variation.

**Table 2 diagnostics-12-01999-t002:** Analytical accuracy of all the methods analysed.

Method/Reagent	% Recovery		
	Lab 1	Lab 2	Lab 3
**OSA/SynthasIL** hs/FVIIIdef	135.4		
**OSA/SynthasIL** hs/FVIIIdef+VWF		97.3	101.6
**OSA/SynthasIL** rpFVIIIs/FVIIIdef	84.6 **	36.2 **	38.6 **
**OSA/SynthasIL** rpFVIIIs/FVIIIdef+VWF	110.0	110.1	113.0
**OSA/SynthaFax** hs/FVIIIdef		60.0	
**OSA/SynthaFax** hs/FVIIIdef+VWF	76.4		79.4
**OSA/SynthaFax** rpFVIIIs/FVIIIdef	65.3	77.0	56.7
**OSA/SynthaFax** rpFVIIIs/FVIIIdef+VWF	118.7 #	182.8 #	138.4 #
**CSA/Coamatic FVIII** hs	62.9	62.3	59.3
**CSA/Coamatic FVIII** rpFVIIIs in FVIIIdef	133.0	124.3	132.0
**CSA/Coamatic FVIII** rpFVIIIs in FVIIIdef+VWF	104.6 °	95.7 °	89.4 °

** *p* < 0.0001 Lab1 vs. Lab 2 and Lab 3, # *p* < 0.0001 Lab 2 vs. Lab 1 and Lab 3 and Lab 1 vs. Lab 3, ° *p* = 0.004 Lab 1; vs. Lab 2 and Lab 3; Abbreviations: OSA = one-stage assay; CSA = chromogenic assay; hs = human standard; FVIIIdef = FVIII deficient plasma; FVIIIdef+VWF = FVIII deficient plasma with von Willebrand Factor; rpFVIIIs = recombinant FVIII porcine standard.

**Table 3 diagnostics-12-01999-t003:** Inter-laboratory variability.

Method/Reagent	CV %	% Recovery
**OSA/SynthasIL** hs/FVIIIdef+VWF	**4.04**	**99.43**
**OSA/SynthasIL** rpFVIIIs/FVIIIdef	46.47	52.14
**OSA/SynthasIL** rpFVIIIs/FVIIIdef+VWF	**2.75**	**111.04**
**OSA/SynthaFax** hs/FVIIIdef+VWF	2.84	77.89
**OSA/SynthaFax** rpFVIIIs/FVIIIdef	13.94	66.32
**OSA/SynthaFax** rpFVIIIs/FVIIIdef+VWF	21.45	146.63
**CSA/Coamatic FVIII** hs	5.12	61.50
**CSA/Coamatic FVIII** rpFVIIIs in FVIIIdef	4.67	129.79
**CSA/Coamatic FVIII** rpFVIIIs in FVIIIdef+VWF	**8.32**	**96.57**

Abbreviations: OSA = one-stage assay; CSA = chromogenic assay; hs = human standard; FVIIIdef = FVIII deficient plasma; FVIIIdef+VWF = FVIII deficient plasma with von Willebrand Factor; rpFVIIIs = recombinant FVIII porcine standard. In bold are the three combinations of reagents presenting satisfactory analytical performance.

## Data Availability

Data available on request.

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
