# Peer review of "Analytical Performance of Different Laboratory Methods for Measuring Susoctocog-Alfa"

_diagnostics, 2022, doi:10.3390/diagnostics12081999_

Round 1
Reviewer 1 Report
The manuscript entitled “Analytical performance of different laboratory methods for measuring susoctocog-alfa’’ by Novembrino et al.,, is a report of a complex study determining the analytical performance of various reagent and calibrator combinations for the purpose of measuring rpFVIII activity in plasma. It is an interesting study, however, I have following queries to be addressed
1. The authors have not provided any reference regarding how and what concentration of different reagents was selected for the study.
2. In the figure and tables, Indicate, using different symbols, if there are significant differences between the analytical performance of different combinations of reagents and calibrators for measuring rpFVIII activity in plasma. Please mention statistical values for greater representation of data.
3. In the discussion section, Organize, and address if the investigation of the analytical conditions to measure rpFVIII activity in plasma overcomes the limitations of commercially available assays
Author Response
Comments and Suggestions for Authors
The manuscript entitled “Analytical performance of different laboratory methods for measuring susoctocog-alfa’’ by Novembrino et al.,, is a report of a complex study determining the analytical performance of various reagent and calibrator combinations for the purpose of measuring rpFVIII activity in plasma. It is an interesting study, however, I have following queries to be addressed
- The authors have not provided any reference regarding how and what concentration of different reagents was selected for the study.
As suggested, the authors added the information required in the Reagents paragraph of the Materials and Methods section
- In the figure and tables, Indicate, using different symbols, if there are significant differences between the analytical performance of different combinations of reagents and calibrators for measuring rpFVIII activity in plasma. Please mention statistical values for greater representation of data.
The authors inserted the results of the statistical analysis in the tables, as suggested, and reported the details in the Statistical analysis paragraph of the Materials and Methods section
- In the discussion section, Organize, and address if the investigation of the analytical conditions to measure rpFVIII activity in plasma overcomes the limitations of commercially available assays
The authors added some sentences regarding this topic in the Conclusions section.
Reviewer 2 Report
Acquired hemophilia (AHA) is an uncommon but severe bleeding disorder, which is caused by the development of autoantibodies directed against endogenous factor VIII, leading to decreased FVIII activity and a bleeding phenotype. Patients who present with major bleeding require initiation of appropriate antihemorrhagic treatment in a timely fashion. Although there is extensive experience with bypassing agents (BPAs) such as recombinant activated FVII or activated prothrombin complex concentrates, both having similar efficacy and side effect profiles, disadvantages to BPAs include the inability to monitor response with standard laboratory tests. A recombinant, B-domain-deleted, porcine FVIII (rpFVIII), was approved by the FDA for treatment of AHA in 2014. rpFVIII is appealing as it can be monitored with one-stage clot-based FVIII assays (OSA). The use of chromogenic assays (CSA) for this purpose is disputed due to high underestimation of FVIII levels. On this basis, the authors of the submitted manuscript aim to evaluate which method and reagents may provide the best analytical performance to measure the level and plasma activity of rpFVIII (Obizur, Baxalta US Inc.-Takeda company). Their study involved three Italian laboratories, testing a total of eleven combinations of reagents with two different assays (OSA and CSA). Intra- and inter-assay and inter-laboratory coefficient of variation were calculated together with percentage of recovery on the expected values.
The submitted manuscript is technically sound and written in a clear and comprehensive manner. However, since there are a few typos (see below), I recommend another round of proof-reading. Furthermore I ask the authors to adapt Figure 1, since it is not yet completely clear to the reader: (1) since you show the mean values, I ask you to also show the standard deviations. (2) For the reader it would be much easier, if you indicate the three shown combinations also in the graphs and not only in the figure legend. (3) I ask you to the change the order of the combinations (a: OSA/SynthasIL: rpFVIIIs/FVIIIdef+vWF; b: OSA/SynthasIL: hs/FVIIIdef+vWF; c: CSA/Coamatic FVIII: rpFVIIIs in FVIIIdef+vWF) according to the ranking in the tables (a: OSA/SynthasIL: hs/FVIIIdef+vWF; b: OSA/SynthasIL: rpFVIIIs/FVIIIdef+vWF; c: CSA/Coamatic FVIII: rpFVIIIs in FVIIIdef+vWF). (4) Please indicate the concentrations for C1-C5 (1.5-0.05) also in the figure legend and write that they are given in IU/mL.
Additionally I ask the authors to state why several combinations (OSA/SynthasIL: hs/FVIIIdef; OSA/SynthasIL: hs/FVIIIdef+vWF; OSA/SynthaFAX: hs/FVIIIdef; OSA/SynthaFax: hs/FVIIIdef+vWF) have been tested only in one or two laboratories, instead of in all three? From my point of view this has to be mentioned in section “5. Limitations” as well.
Although the manuscript focus on laboratory methods, some clinical information related to the treatment of AHA is given. Emicizumab is of great interest in this setting. However onset of hemostatic effect is delayed and this fact should be mentioned. Tiede A. and Knoebl P. have published essential information on this topic and should be referenced. Kruse et al are planning/ have started a clinical trial on the use of Emicizumab in AHA. Authors are advised to clarify statement on the use of Emicizumab in AHA.
Minor changes:
- Page 1, lane 26: a space character is missing after Boston
- Page 1, lane 27: delete the hyphen after VWF
- Page1, lane 42: since “neutralise” is British English, I recommend to write “neutralize”
- Page 1, lane 42: consistency: in the abstract the authors write “factor VIII”, here “Factor VIII”- I recommend to write the latter also in lowercase
- Page 2, lane 48-56: please split this sentence since it is rather too long
- Page 2, lane 57-61: please split this sentence
- Page 2, lane 62: considerable (instead of considerably) side effects
- Page 2, lane 95: presented (instead of present) study
- Page 3, lane 133-134: Sentence incomplete: In Table 1 inter- and intra-assay imprecision obtained for each of the three laboratories (is shown).
- Page 4, Table 1: consistency: check the space characters of the combinations in the method/reagent column
- Page 5, lane 158: consistency: the authors here write “table”, in the remaining text “Table”
- Page 6, lane 176: b) SynthasIL with (instead “and”) FVIII deficient plasma
- Page 8, lane 181-185: please split this sentence since it is rather too long
- Page 8, lane 216: presented (instead of present) study
- Page 9, lane 232: please use “patient plasma” instead of “plasma patient”
- Page 9, lane 243: “it” missing after “make”
- Page 9, lane 252: comma missing after “evaluated”
Author Response
Comments and Suggestions for Authors
The submitted manuscript is technically sound and written in a clear and comprehensive manner. However, since there are a few typos (see below), I recommend another round of proof-reading. Furthermore I ask the authors to adapt Figure 1, since it is not yet completely clear to the reader: (1) since you show the mean values, I ask you to also show the standard deviations. (2) For the reader it would be much easier, if you indicate the three shown combinations also in the graphs and not only in the figure legend. (3) I ask you to the change the order of the combinations (a: OSA/SynthasIL: rpFVIIIs/FVIIIdef+vWF; b: OSA/SynthasIL: hs/FVIIIdef+vWF; c: CSA/Coamatic FVIII: rpFVIIIs in FVIIIdef+vWF) according to the ranking in the tables (a: OSA/SynthasIL: hs/FVIIIdef+vWF; b: OSA/SynthasIL: rpFVIIIs/FVIIIdef+vWF; c: CSA/Coamatic FVIII: rpFVIIIs in FVIIIdef+vWF). (4) Please indicate the concentrations for C1-C5 (1.5-0.05) also in the figure legend and write that they are given in IU/mL.
Figure 1 has been revised as suggested by the reviewer. Standard deviations were not visible in the figure because their very low values are incompatible with the graphic scale.
Additionally I ask the authors to state why several combinations (OSA/SynthasIL: hs/FVIIIdef; OSA/SynthasIL: hs/FVIIIdef+vWF; OSA/SynthaFAX: hs/FVIIIdef; OSA/SynthaFax: hs/FVIIIdef+vWF) have been tested only in one or two laboratories, instead of in all three? From my point of view this has to be mentioned in section “5. Limitations” as well.
As suggested, a sentence explaining the lack of some results has been added in the Limitations paragraph.
Although the manuscript focus on laboratory methods, some clinical information related to the treatment of AHA is given. Emicizumab is of great interest in this setting. However onset of hemostatic effect is delayed and this fact should be mentioned. Tiede A. and Knoebl P. have published essential information on this topic and should be referenced. Kruse et al are planning/ have started a clinical trial on the use of Emicizumab in AHA.
As suggested, the authors addressed this topic in the Discussion section.
Minor changes:
- Page 1, lane 26: a space character is missing after Boston
- Page 1, lane 27: delete the hyphen after VWF
- Page1, lane 42: since “neutralise” is British English, I recommend to write “neutralize”
- Page 1, lane 42: consistency: in the abstract the authors write “factor VIII”, here “Factor VIII”- I recommend to write the latter also in lowercase
- Page 2, lane 48-56: please split this sentence since it is rather too long
- Page 2, lane 57-61: please split this sentence
- Page 2, lane 62: considerable (instead of considerably) side effects
- Page 2, lane 95: presented (instead of present) study
- Page 3, lane 133-134: Sentence incomplete: In Table 1 inter- and intra-assay imprecision obtained for each of the three laboratories (is shown).
- Page 4, Table 1: consistency: check the space characters of the combinations in the method/reagent column
- Page 5, lane 158: consistency: the authors here write “table”, in the remaining text “Table”
- Page 6, lane 176: b) SynthasIL with (instead “and”) FVIII deficient plasma
- Page 8, lane 181-185: please split this sentence since it is rather too long
- Page 8, lane 216: presented (instead of present) study
- Page 9, lane 232: please use “patient plasma” instead of “plasma patient”
- Page 9, lane 243: “it” missing after “make”
- Page 9, lane 252: comma missing after “evaluated”
All the typing errors has been corrected